# Serum BDNF’s Role as a Biomarker for Motor Training in the Context of AR-Based Rehabilitation after Ischemic Stroke

**DOI:** 10.3390/brainsci10090623

**Published:** 2020-09-09

**Authors:** Ekaterina S. Koroleva, Ivan V. Tolmachev, Valentina M. Alifirova, Anastasiia S. Boiko, Lyudmila A. Levchuk, Anton J. M. Loonen, Svetlana A. Ivanova

**Affiliations:** 1Department of Neurology and Neurosurgery, Siberian State Medical University, Moskovsky trakt, 2, 634050 Tomsk, Russia; kattorina@list.ru (E.S.K.); v_alifirova@mail.ru (V.M.A.); 2Department of Medical and Biological Cybernetics, Siberian State Medical University, Moskovsky trakt, 2, 634050 Tomsk, Russia; ivantolm@mail.ru; 3Mental Health Research Institute, Tomsk National Research Medical Center of the Russian Academy of Sciences, Aleutskaya str., 4, 634014 Tomsk, Russia; anastasya-iv@yandex.ru (A.S.B.); rla2003@list.ru (L.A.L.); ivanovaniipz@gmail.com (S.A.I.); 4PharmacoTherapy, -Epidemiology and -Economics, Groningen Research Institute of Pharmacy, University of Groningen, Antonius Deusinglaan 1, 9713AV Groningen, The Netherlands; 5Department of Psychiatry, Addictology and Psychotherapy, Siberian State Medical University, Moskovsky trakt, 2, 634050 Tomsk, Russia

**Keywords:** BDNF, ischemic stroke, rehabilitation, augmented reality (AR)-biofeedback motion training, long-term potentiation, functional rewiring

## Abstract

Background: brain-derived neurotrophic factor (BDNF) may play a role during neurorehabilitation following ischemic stroke. This study aimed to elucidate the possible role of BDNF during early recovery from ischemic stroke assisted by motor training. Methods: fifty patients were included after acute recovery from ischemic stroke: 21 first received classical rehabilitation followed by ‘motor rehabilitation using motion sensors and augmented reality’ (AR-rehabilitation), 14 only received AR-rehabilitation, and 15 were only observed. Serum BDNF levels were measured on the first day of stroke, on the 14th day, before AR-based rehabilitation (median, 45th day), and after the AR-based rehabilitation (median, 82nd day). Motor impairment was quantified clinically using the Fugl–Meyer scale (FMA); functional disability and activities of daily living (ADL) were measured using the Modified Rankin Scale (mRS). For comparison, serum BDNF was measured in 50 healthy individuals. Results: BDNF levels were found to significantly increase during the phase with AR-based rehabilitation. The pattern of the sequentially measured BDNF levels was similar in the treated patients. Untreated patients had significantly lower BDNF levels at the endpoint. Conclusions: the fluctuations of BDNF levels are not consistently related to motor improvement but seem to react to active treatment. Without active rehabilitation treatment, BDNF tends to decrease.

## 1. Introduction

In 2015, neurological disorders accounted for 10.2% of lost disability-adjusted life-years (DALYs) and 16.8% of deaths globally [1], making them the leading and second-leading causes of DALY loss and death, respectively [1,2]. Strokes accounted for 47.3% of the DALY loss and 67.3% of the deaths attributable to neurological disorders as a whole [1]. According to the Global Burden of Disease (GBD) 2016 study, the lifetime risk of stroke from the age of 25 years was 24.9%–18.3% for ischemic stroke and 8.2% for hemorrhagic stroke [3]. In Russia, the mean lifetime risk was slightly higher (32.8% for both sexes combined) [2]. In 2017, the age-standardized ischemic stroke incidence, DALY loss and death rates in the Russian federation were 139.7 (126.0–155.6), 93.0 (91.4–96.0), and 1571.0 (1483.1–1654.6) (rates per 100,000 (95% CI)), respectively [4].

Stroke causes motor-function deficits that significantly reduce the patient’s mobility, activities of daily living (ADL), and participation in social activities, resulting in a decreased quality of life (QoL) [5]. Training-based rehabilitation is effective but requires the proper resources and specialized equipment [6]. Virtual Reality (VR)-based rehabilitation is an economical and safe method for improving motor and cognitive functions, facilitating brain plasticity, and regenerative processes [7]. It is based on training regarding the integration of the sensory information supplied by visual, auditory, tactile, and somatosensory perception during the execution of motor activities [8,9]. Han Suk Lee and colleagues have shown that adding VR-based training to standard treatment improves the gait, balance, lower limb movement, lower limb strength, and lower limb muscle tone of patients with chronic stroke [10]. A systematic literature review and meta-analysis showed that the VR-based rehabilitation of patients with stroke was more effective than conventional treatment for improving ADL and upper limb function [5,11]. Vourvopoulos et al. showed beneficial effects of EEG-based VR and brain–computer interfaces (BCIs) in patients with severe motor impairments [12].

Brain-derived neurotrophic factor (BDNF) is one of the most extensively studied neurotrophic factors of the mammalian brain. It was discovered in 1982 by Barde and colleagues [13] and belongs to the neurotrophin family comprising BDNF, nerve growth factor (NGF), neurotrophin-3 (NT-3), and neurotrophin-4 (NT-4) [14]. BDNF is involved in neuronal growth and development, neuroprotection, and, particularly, the short- and long-term adaptation of synaptic activity [15,16]. Mature BDNF binds to the tropomyosin-related kinase B (TrkB, tyrosine kinase B) receptor and triggers signal transduction cascades (Insulin Receptor Substrate 1 or 2 (IRS1/2), Phosphoinositide 3-kinases (PI3Ks), and Protein kinase B (PKB or Akt), resulting in protein synthesis, axonal growth, dendritic maturation, synaptic plasticity, stress resistance, cell survival, and therapeutic neovascularization. However, pro-BDNF has dissimilar effects. By activating the common low-affinity nerve growth factor receptor (p75 neurotrophin receptor), pro-BDNF induces the apoptosis of endothelial cells as well as vascular smooth muscles and impairs angiogenesis [17,18]. Neuronal remodeling in the cerebral cortex and of its connectivity to the striatum is one of the components of self-healing following ischemic stroke and provides compensatory (restorative) brain function. BDNF is well-known for its neuroprotective effects [19] but also plays an essential role in mediating the sustained enhancement of the excitatory synaptic strength of glutamatergic synapses, i.e., long-term potentiation (LTP) [20,21]. This is usually associated with hippocampal synapses, with a role in episodic memory [20,21], but also plays a role in other cortical areas [22], including the primary motor cortex [23].

The aim of this study was to explore the putative role of BDNF in rehabilitation following ischemic stroke—specifically, to assess the dynamics of serum BDNF induction by the stimulation of the motor cortex by Augmented Reality (AR)-biofeedback motion training in the early recovery period.

## 2. Experimental Section

### 2.1. Studied Patients

Thirty-five ischemic stroke patients over the age of 18 (20 men and 15 women) who had received AR-based rehabilitation from January to December 2019 at the Siberian State Medical University (SSMU), Tomsk, Russia, were recruited. The study was approved by the Ethics Committee of the SSMU, Russia (protocol no. 5961; 18 June 2018), and all participants gave informed consent. Patients with first-in-life stroke, as clinically manifested and identified according to the WHO clinical criteria, were included. Ischemic stroke of the middle cerebral artery was confirmed by Magnetic Resonance Imaging (MRI) or computerized tomography (CT) scans. Patients with previous neurological disease, chronic degenerative or inflammatory diseases, or malignancies were not included.

Patients were recruited for participation after discharge from the Tomsk Regional Stroke Center (Tomsk RSC) (Days 14–16) and received AR-based rehabilitation during a 10 day course in the last part of the early recovery period (Days 60–90).

The inclusion criteria for motor rehabilitation using motion sensors and augmented reality (AR-based rehabilitation) were unilateral stroke; an age between 18 and 80 years; mild-to-moderate upper limb hemiparesis (National Institutes of Health Stroke Scale, NIHSS < 12) [24]; a Modified Rankin Scale (mRS) score < 4 [25]; a Modified Ashworth scale (MAS) score ≤ 2 [26,27]; and the absence of any cognitive impairment (Mini-Mental State Evaluation (MMSE) ≤ 22) [28] or behavioral dysfunction. The exclusion criteria were bilateral impairment, severe sensory deficits in the paretic upper limb, a loss of sense of the positions of muscles and joints, other damage to the central and peripheral neural systems, and a patient’s refusal to participate in this part of study.

### 2.2. Healthy Individuals

Blood samples were collected by antecubital venipuncture from 50 individuals with no history of stroke, and the composition of this group was comparable to that of the stroke patients with respect to gender, age, and the presence of risk factors. The patients were examined in an outpatient unit.

### 2.3. Study Design

The study population was categorized into two groups. The group A-Rehab-Traditional-AR included 21 patients who received a 28 day rehabilitation course involving traditional exercise programs in the Siberian Federal Scientific Clinical Center of the Federal Medical-Biological Agency (within 16–60 days of the stroke) before AR-based rehabilitation (within 60–90 days). The group B-Rehab-AR consisted of 14 patients who only received AR-based rehabilitation within the early recovery period (Days 60–90). The group C-outpatient observation (OPO) comprised 15 patients who were under outpatient observation (OPO) after an acute period of ischemic stroke and did not receive any rehabilitation treatment. Details of the applied rehabilitation techniques are provided in Appendix B, Appendix C and Appendix D. Serum BDNF concentrations were assessed before and after the AR-based rehabilitation in the patients of the groups A-Rehab-Traditional-AR and B-Rehab-AR, and before and after outpatient observation in the patients of C-OPO over the same time period. These levels were compared to those of the control group. We considered BDNF levels at four specific time points in order to follow their course throughout the acute and early recovery periods (the study design is illustrated in Appendix A):Point 1—first day of stroke.Point 2—discharge from the Tomsk RSC (median, 14th day; range, 14th–16th day).Point 3—after traditional rehabilitation/before AR-based rehabilitation (median, 45th day; range, 16th–60th day) (not in the group C-OPO).Point 4—after AR-based rehabilitation (median, 82nd day; range, 60th–90th day).

### 2.4. Exercise Training and Experimental Procedure

To stimulate the motor cortex, we chose a method of influencing sensory systems using biological feedback. Sensory cortex stimulation was performed using augmented reality, which is good in terms of patient safety [29]. We created a controlled virtual environment that visualizes sensory stimuli, the characteristics of which depend on the parameters of human movement. Studies have shown virtual reality might be a promising approach to enriching and improving gait rehabilitation after stroke [30]. In our case, we used a more gentle impact method to generate the feedback stimuli.

In order to assess the impact of visual stimuli on motor function, we developed specialized software (NeuroRAR—software for the motor rehabilitation of neurological patients using motion sensors and augmented reality; certificate of state registration of a computer program no. 2019619570; dated 19 July 2019).

The program consists of three modules: 1. RehabDT, which collects and analyses data from video capture sensors on the patient’s body; 2. RehabAR, which visualizes the obtained data for the patient on augmented reality glasses and analyzes the motor quality within the Accuracy, Statics, Capture and Balance Domains; and 3. RehabCS, which processes and stores the motion capture data. Details are provided in Appendix C.

The AR protocol involved a 10-day course of motor-training sessions held on a consecutive daily basis once in the morning. The time to complete one motor domain was 20 min. The rest time between tasks within the domain, at the request of the patient, was 1–2 min; that between the four motor tasks was 5 min. The patients followed the augmented reality (AR)-based training protocol under the supervision of a kinesiotherapist.

### 2.5. BDNF Assessment

After patients provided informed consent, blood was collected into BD Vacutainer^®^ tubes with a clot activator (CAT) to isolate the serum (BD, Franklin Lakes, NJ, USA), which was separated by centrifugation at 2000 RCF at 4 °C for 20 min. The serum samples were stored at −80 °C until analysis. The mature BDNF concentration was determined with the MAGPIX multiplex analyzer (Luminex, Austin, TX, USA) using xMAP^®^ Technology in the Mental Health Research Institute based at The Core Facility “Medical Genomics”, Tomsk NRMC. The collected information was processed by the Luminex xPONENT^®^ software, with the subsequent export of data to MILLIPLEX^®^ Analyst 5.1.

### 2.6. Clinical Assessment of Motor Functions

Essential patient characteristics were obtained from the patients’ medical files and/or standard clinical interviews. We used the clinical evaluation of motor impairment and Activities of Daily Living (ADL) as the primary outcome of this study into AR-based rehabilitation. These were measured on two standardized clinical-rating scales: the Fugl–Meyer scale (FMA) was used as an instrument for quantitatively evaluating post-stroke motor disorders [31,32], and the Modified Rankin Scale (mRS) measured functional disability and ADL [23].

Gait and movements of upper extremity changes during AR-rehabilitation were assessed using author parameters’ quality of movements:Variability of movements when following a given trajectory;Total number of completed movements (completed task) during one motor session;Maximum duration of a tasks series in one approach without a rest;Variance of the displacement for the central point of the body during the walk before crossing the obstacles;Height of raising the leg on the affected side when stepping through virtual barrier.

The direct effects of AR-based rehabilitation were also evaluated by calculating the quality of movement during the first, fifth, and tenth training sessions, and interpreting detailed information obtained with parameters for assessing the quality of movements during the tasks in motor domains [33].

### 2.7. Statistics

All data was analyzed with the SPSS statistical package (20.0 Version). The results are expressed as percentages for categorical variables or as medians (Me) and interquartile ranges (IQR). The Shapiro–Wilk test was used to analyze the normality of distribution of the variables (*p* > 0.05). Quantitative data without a normal distribution were analyzed with non-parametric tests. The chi-square (χ^2^) test was used to assess the differences in the sizes of distributions of qualitative data. Differences in the parameters between groups were assessed with the non-parametric Mann–Whitney U test; for related samples, we used the Wilcoxon Rank-Sum test. The Bonferroni test was used as a lead for multiple post hoc comparisons. A *p* value less than 0.05 was considered statistically significant.

The differences between serum BDNF at Points 2 and 4 (ΔBDNFobserved = BDNF day82 − BDNF day14) and that at Points 3 and 4 (ΔBDNFobserved = BDNF day82 − BDNF day45) were used to estimate the synaptic-plasticity-related and activity-dependent increases in BDNF during the motor training. The difference between the initial and 14-day serum BDNF (ΔBDNFobserved = BDNF day14 − BDNF initial) was used to estimate the peak neuroplasticity in response to ischemic brain damage (spontaneous plasticity).

## 3. Results

Fifty ischemic stroke patients and an equal number of healthy individuals were recruited in this study. Their demographic data are presented in Appendix E. Age, gender, height, weight, and risk factors for developing cerebrovascular diseases did not significantly differ between them or with and between the relevant subgroups of patients and controls. The detailed characteristics of the patient groups—including those according to the criteria of Trial of Org 10,172 in Acute Stroke Treatment (TOAST) [34]—are described by Koroleva and colleagues [33].

### 3.1. BDNF Levels

The serum BDNF levels measured in the patients from the groups A-Rehab-Traditional-AR, B-Rehab-AR, and C-OPO are shown in Figure 1 and Table 1 and Table 2.

In Figure 1 and Table 2, it is apparent that the serum BDNF levels peaked at time point 2 on Day 14 of the stroke in A-Rehab-Traditional-AR, B-Rehab-AR, and C-OPO patients (ΔBDNF day14−initial in Table 1), directly after the first study phase following the ischemic stroke incident. A comparison of the BDNF levels and their change at/between time points 2 and 4 (ΔBDNF day82−day14) shows similar decreases in the patients of all three groups. Notably, the BDNF levels on Day 82 were decreased to a major extent in group C-OPO patients, who did not receive any rehabilitation programs during the early recovery period (Probability Point 2 vs. Point 4 (P_2-4_) < 0.001, Figure 1; Probability A-Rehab-Traditional-AR vs. C-OPO (P_A-C_) = 0.049, Probability B-Rehab-AR vs. C-OPO (P_B-C_) = 0.021, Table 1).

It should be noted that the absolute BDNF level fluctuations between Points 2 and 3 were dissimilar between the patients of A-Rehab-Traditional-AR, who received active treatment with traditional rehabilitation approaches, and B-Rehab-AR, who were under outpatient observation during this time (Figure 1). Although the decrease in serum BDNF levels was significant in A-Rehab-Traditional-AR (Probability Point 2 vs. Point 3 (P_2-3_) = 0.010) as well as in B-Rehab-AR (P_2-3_ = 0.016), they were significantly lower at the end of this period (Probability A-Rehab-Traditional-AR vs. B-Rehab-AR (P_A-B_) = 0.049 at Point 3) in the last patient group. This indicates that the BDNF levels decreased less in patients who received active treatment during this period.

In both A-Rehab-Traditional-AR and B-Rehab-AR, the BDNF levels significantly increased during AR rehabilitation (Probability Point 3 vs. Point 4 (P_3-4_) = 0.012 and P_3-4_ = 0.017, accordingly). An increase in serum BDNF levels during the activity-dependent stimulation of the motor cortex by the AR-biofeedback-motion training in A-Rehab-Traditional-AR and B-Rehab-AR is also observed when considering the increase in levels before and after AR-based rehabilitation (ΔBDNF Day 82–Day 45 in Table 1). The increase in serum BDNF levels tended to be significantly higher between the 45th and 82nd days of observation in B-Rehab-AR (observation period between Points 2 and 3) than in A-Rehab-Traditional-AR (P_A-B_ = 0.049, Table 1). In Figure 1, we can see that the levels of BDNF after the AR-based rehabilitation in A-Rehab-Traditional-AR and B-Rehab-AR reached the same levels as those at Point 2 but did not exceed them and were statistically indistinguishable from them (P_2-4_ = 0.476 and P_2-4_ = 0.374, accordingly) and each other (P_A-B_ = 0.10 in Point 4).

### 3.2. Clinical Assessment of Motor Recovery

The analysis of the FMA scores for the stroke patients at the observation points is presented in Figure 1 and Table 3. The changes in mRS scores during the AR-based rehabilitation are shown in Figure 2.

The analysis of motor recovery according to FMA demonstrated similar and significant improvements of motor functions in the upper and low extremities and balance in the three groups at Day 14 (Figure 1, Table 3). However, A-Rehab-Traditional-AR patients had significantly lower total FMA scores at Point 2 than patients of B-Rehab-AR (P_A-B_ = 0.01) (Figure 1). In terms of the relationship with serum BDNF levels, we found that the initial BDNF levels in the 21 patients of A-Rehab-Traditional-AR were consistently lower than those in the healthy subjects (P_A-Ctrl_ = 0.001 at Point 1) and not significantly different to those in B-Rehab-AR (P_A-B_ = 0.21 at Point 1). A comparison between the BDNF levels as measured in 50 ischemic stroke patients and the same number of comparable healthy individuals is shown in Table 2 and illustrated in Figure 1. No significant clinical differences were observed at entry between the three groups of patients (Figure 1). In spite of these consistently lower BDNF levels compared with those of the control group at all time points, the motor handicap as quantified by FMA gradually improved in A-Rehab-Traditional-AR (Figure 1). The traditional rehabilitation performed in A-Rehab-Traditional-AR patients during the second phase of the study allowed the equalization of the motor handicap between the two groups at Point 3 (Figure 1). The highly significant improvement of total FMA scores observed in A-Rehab-Traditional-AR (P_2-3_ < 0.001) was absent in B-Rehab-AR (P_2-3_ = 0.30), as shown in Figure 1. This large improvement of motor and ADL impairment was accompanied by a significant decrease in peripheral BDNF levels (Figure 1), but this decrease was even larger in B-Rehab-AR. The BDNF levels measured in our patients appear to also correspond to a process other than neuroplastic changes within motor circuits. We also noticed that the balance function measured by FMA-Balance did not improve during the traditional rehabilitation program in A-Rehab-Traditional-AR (P_2-3_ = 0.06, Table 3).

During AR-based rehabilitation treatment, the changes in serum BDNF levels were accompanied by an improvement of motor function, as quantified with the Fugl–Meyer rating scale (FMA) (Figure 1), and a reduction in the degree of functional disability, as measured by the mRS (Figure 2). The amount of improvement and endpoint total FMA scores were not significantly different (P_A-B_ = 0.45), despite significantly lower values of ΔBDNF Day 82–Day 45 and the level of BDNF at Point 4 in A-Rehab-Traditional-AR being significantly lower than that in B-Rehab-AR (P_A-B_ = 0.02). A detailed analysis of the FMA scores revealed a number of features as shown in Table 3. One of them was a lack of significant recovery of the lower extremities during AR-biofeedback-motion training in B-Rehab-AR, as assessed by FMA-Low extremity (P_3-4_ = 0.080). However, balance function in A-Rehab-Traditional-AR, which did not recover well after traditional approaches, showed a significant improvement according to FMA-Balance (P_2-3_ = 0.06 and P_3-4_ = 0.002, accordingly).

The clinical assessment of motor recovery in C-OPO at Point 4 found no significant change in clinical condition according to total FMA score from the 14th day to the end of the early recovery period (P_2-4_ = 0.750, Figure 1). Both the measured peripheral BDNF levels and the motor/ADL function scores were significantly lower in C-OPO (who did not receive any active motor rehabilitation during the early recovery period) than those in A-Rehab-Traditional-AR and B-Rehab-AR at the end of the trial (Figure 1).

Comparison of the movement parameters concerning the variability of movements during the 1–10 training sessions did not reveal any significant differences between A-Rehab-Traditional-AR and B-Rehab-AR [33]. During AR-rehabilitation significant augmentation of the accuracy of following a given trajectory, improvement of reciprocal interaction of upper extremities muscles with a static-dynamic load and increased the muscle strength was observed (Figure 3). Patients also significantly increased the height of raising their leg on the affected side since 1st to 10th sessions of AR-rehabilitation (Figure 4).

## 4. Discussion

In the present study, we tried to elucidate the relationship between serum BDNF levels and motor recovery in response to AR-based rehabilitation over a 60–90-day post-stroke period, which is still considered to belong to the plastic window for stroke recovery [35,36]. Our results show that AR-based rehabilitation is accompanied by a significant augmentation of serum BDNF levels as well as by a highly significant improvement of motor function. Without treatment, the BDNF levels spontaneously decrease after the acute phase and the clinical condition is more or less stable. Throughout the 3-month recovery period, the serum BDNF levels in the 21 patients who received both classical and AR-based rehabilitation treatment were still significantly lower than those in the 50 healthy controls. Although interesting from a neuroplasticity perspective, we decided to exclude the absolute change during the first 14 days following stroke from further consideration. Peripheral BDNF is largely concentrated within thrombocytes, from where it is released by thrombocyte activation during clotting [37]; this also explains why serum levels are far higher than plasma levels [38]. The disequilibrium induced by acute cerebrovascular pathology during the first few days may reduce the relationship of serum BDNF with specific pathophysiological CNS processes. The possible existence of a disequilibrium state during the initial few days may also explain the conflicting results obtained with respect to the relationship between initial BDNF levels and outcomes. Wang et al. found evidence for an association between BDNF and outcomes after 3 months [39], but the findings of others were negative for outcomes after 90 days [40,41,42] and 12 months [42].

After ischemic stroke, patients show spontaneous motor recovery, which is most profound over the first 30 days but continues over the subsequent 30–90 days [36]. There are three suggested reasons for this: 1. the resolution of the acute metabolic depression, which had induced neuronal dysfunction; 2. compensation by the application of other muscle groups to execute the motor tasks; and 3. the local and sometimes distant rewiring of the central nervous system [43]. BDNF may have a role in all three processes. BDNF exerts neuroprotective activities [17], which might protect against direct tissue loss resulting from ischemia in the acute phase of stroke and enable the later recovery of cell function [44]. It plays an essential role in regulating long-term potentiation (LTP) at glutamatergic synapses [21,45,46], which can be considered essential for adapting the activity of intracortical and corticofugal excitatory pathways necessary for the proper execution of motor tasks. Although especially late phase LTP (i.e., synaptic consolidation) is based on structural synaptic adaptations [21,45,46,47], this process is believed to only result in the functional rewiring of existing networks and circuits. The results of Qin and colleagues in a mouse model suggest that local and distant rewiring may be oppositely affected by a knocked-in variant of the *BDNF* gene, which suggest that BDNF primarily shapes the functioning of local ipsilesional circuits [48]. Finally, BDNF has neuroprotective and growth-promoting effects at least on the level of the brainstem and spinal cord, although descending corticospinal axons are predominantly affected by the related factor neurotrophin-3 (NT-3) [49]. Lu et al. observed the expression of BDNF receptors (tyrosine kinase B; TrkB) on the cell bodies and apical dendrites of corticospinal neurons but not on their projecting neurons [49,50]. The observation that BDNF promotes corticospinal motor neuron survival but not the outgrowth of lesioned corticofugal central motor neurons [50], suggests it is unlikely that BDNF has a prominent role in the physical rewiring of neuronal networks by sprouting, neuronal growth, and synaptogenesis [43]. Of note, these last structural changes may depend upon the function of microglia [51]. BDNF may, however, have a role in the structural reorganization by such processes of the dendrites of cortical pyramidal cells, consolidating a changed sensitivity to afferent fibers [52].

The ischemic damage to brain tissue and destruction of neural networks probably induces a reorganization of the neuronal connections within the motor cortex [53,54]. Spontaneous and activity-induced BDNF release contribute to the regulation of this reorganization, and motor recovery is known to depend on the methods used for the stimulation of the motor cortex, excitatory neurotransmission, activity-dependent BDNF release, and the phosphorylation of TrkB receptors [55]. Neurorehabilitation involving training in the proper execution of motor tasks can be hypothesized to intensify spontaneous recovery due to the aforementioned functional rewiring of neuronal networks and circuits, which would relate this process to the influence of BDNF on neuronal functioning. BDNF is secreted in response to neuronal activity induced by acute and long-term exercise, which may play an important role in inducing cortical neuroplastic changes [55,56,57]. In a mouse model of ischemic stroke, neuronal activation with optogenetics resulted in a significant improvement in cerebral blood flow, as well as an increased expression of activity-dependent neurotrophins, including BDNF, in the contralesional cerebral cortex [58]. However, the findings of Qin et al. suggest that BDNF may not play a dominant role there [48]. Still, the repetition of specific tasks during training may also, by itself, increase the neuronal expression of BDNF, facilitating LTP within the involved pathways and circuits. The interaction of mature BDNF and TrkB leads to the sprouting of neural networks, neuronal growth, synaptogenesis, and, probably, motor recovery in our patients. The influence of motor rehabilitation is related to systematic and long-term training in making movements [59,60]. The main cause of neurological rehabilitation is the principle of motor learning, which is intricately related to repetition. Motor learning is essentially motor adaptation, which can occur within minutes or hours. Stopping the training or changing the conditions of their execution leads to the “forgetting” of these adaptations after a relatively short period of time. In order to consolidate the motor program within memory, the exercise must be repeated at least 400 times. Therefore, the results of motor training programs are related to the duration, intensity, and regularity of exercise [61,62]. High levels of reactive oxygen species are produced during extreme exercises, leading to oxidative damage and increased cellular mortality [63]. At the same time, the increase in BDNF expression is linked to the regularity of motor training sessions but not intensity of exercise [64]. Therefore, stroke patients in our study received motor sessions with moderate physical activity on a daily basis. Our study demonstrates a lack of spontaneous recovery during the “plastic window” period in the absence of recovery treatment without rehabilitation treatment in Groups B (16–60 days) and C (16–90 days), as reflected by the absence of any improvements according to clinical and functional parameters, and lower levels of serum BDNF.

In individual assessments of the movements of the paralyzed limb, we found significant improvement of their quality. This improvement of motor control is considered to represent a beneficial effect of the AR-rehabilitation program. However, along with the obvious motor recovery, we also found compensatory mechanisms primarily concerning the balance function. The stability in an upright position was achieved by sufficient support by the affected leg. Stroke patients reduced the height of raising their leg during walking on the spot on a flat horizontal surface during first days of training which resulted in decreasing the dispersion of the movements of the central point of the body [33]. The results of our study suggest the occurring of structural and cellular changes associated with structural/synaptic plasticity in the non-injured hemisphere during the early recovery period.

The members of A-Rehab-Traditional-AR achieved the same level of motor recovery and functional outcomes at the end of study as those of B-Rehab-AR. However, serum BDNF levels in B-Rehab-AR were higher than those in A-Rehab-Traditional-AR at Point 4, despite the lack of traditional rehabilitation between Points 2 and 3. Therefore, the results of our study reveal clinical and laboratory benefits for modern rehabilitation approaches. For more details about motor recovery, we refer to Korelova et al. [33]. During the acute period of stroke, every patient received early rehabilitation within the Tomsk regional vascular center [65]. Motor rehabilitation was accomplished by using passive–active cyclic robotic electromechanical technologies (Armeo Power Hocoma robotic arm exoskeleton, MOTOmed Viva2 simulator) [66]. It is therefore probable that an activity-dependent increase in BDNF release occurred during this acute phase, augmenting the still-lowered serum BDNF levels. Interestingly, inducing a stroke lesion by itself results in reduced BDNF expression [58]. This corresponds to the lowering of the blood BDNF levels in our patients, which stayed significantly lower than those of the healthy controls throughout study in our largest patient group. A subsequent increase due to spontaneous recovery may have contributed to our findings [43], but the levels measured in the control patients who were under outpatient observation only (C-OPO) do not suggest a major influence during the 14–82-day time period. Ischemic stroke induces the upregulation of BDNF and its receptor’s expression [67,68]. BDNF plays an important role in neural survival in the “penumbra” zone and reducing the infarct volume. Animal studies have shown that it promotes anti-apoptotic pathways and promotes the migration of neuroblasts to ischemic areas [69,70]. The comparative assessment of our patient groups confirmed the importance of a specific revalidation program even after the initial two weeks. Patients of C-OPO, who were under outpatient observation after Acute Phase I of the program, had significantly lower serum BDNF and worse motor disorders on the 90th day of stroke than those of A-Rehab-Traditional-AR and B-Rehab-AR. We also observed a statistically significant increase in serum BDNF levels and clinical improvement after AR-rehabilitation in A-Rehab-Traditional-AR and B-Rehab-AR. The aerobic physical training—based on the principle of motor learning as a result of repeated motor exercises with the generation of a biofeedback stimulus—applied in our study apparently led to increased BDNF expression. This corresponds to the observation that endurance exercise induces BDNF expression in the hippocampus [71,72]. By facilitating the LTP of excitatory intracortical and corticostriatal glutamatergic synapses, BDNF would help to consolidate the trained motor programs. It should be emphasized that the learning of complex movements is not so much associated with a structural reorganization of existing and the creation of new neuronal networks but probably mainly achieved by adapting the activity of extrapyramidal neuronal circuits during the process of motor learning [73,74]. The adaptation of the sensitivity of corticostriatal glutamatergic synapses by long-term potentiation (LTP) and long-term depression (LTD) may similarly induce the functional remodeling of neuronal connectivity within the cerebral cortex. Acetylcholine plays a crucial role in this process, when released both from cholinergic interneurons within the dorsal striatum and from basal forebrain projections [74,75]. Post-stroke functional remodeling by LTP/LTD and structural reorganization resulting from neuroplasticity are considered complementary processes. Cerebral reorganization undoubtedly contributes to functional recovery after stroke. We therefore focused not only on the principle of multiple repetitions, but also on complex movements in AR-rehabilitation-motor-training domains. Although motor function as was reflected by FMA and mRS scores significantly improved in patients who received active rehabilitation treatment, serum BDNF levels are probably not directly related to their clinical condition. This may be caused by the dependence of motor function on different neuroplastic processes simultaneously and because BDNF’s role in adapting motor function may be partly opposite in different brain structures [48].

BDNF has been shown to be involved in various brain pathologies and mental disorders. This includes post-stroke depression, which is a common complication of stroke and may impair the outcomes of rehabilitation [76], such as by affecting motivation. We did not test patients with formal rating scales for anxiety or depression in our study, but they were examined by a multidisciplinary team including a psychiatrist when they were discharged from the vascular center. The rehabilitation prognosis was determined to be favorable in this sample, and they did not have severe post-stroke depression.

Our study has several limitations. The number of participants was rather low, particularly those in the groups B-Rehab-AR and C-OPO. This would have increased the influence of individual response types and decreased the likelihood of finding significant differences. Moreover, we only measured BDNF levels and only after the ischemic stroke incident had occurred. Lower levels of BDNF are correlated with an increased risk of stroke and a poorer prognosis [66,69,77], and hence, the premorbid BDNF levels in the patients may have been (somewhat) lower than the levels in the healthy controls.

## 5. Conclusions

Ischemic stroke results in a reduction of peripheral BDNF levels, which is partly compensated by rehabilitation treatment. AR-based rehabilitation also improves the clinical condition. Without treatment, serum BDNF levels fall significantly and the patients show no clinical or functional improvement. The fluctuations of the serum BDNF levels are independent of the actual clinical condition of the patients. We hypothesize that the BDNF levels depend upon at least two independent processes. Active training in motor tasks is accompanied by an increase of the serum BDNF levels, which may be related to the consolidation of the learned motor skills.

## Figures and Tables

**Figure 1 brainsci-10-00623-f001:**
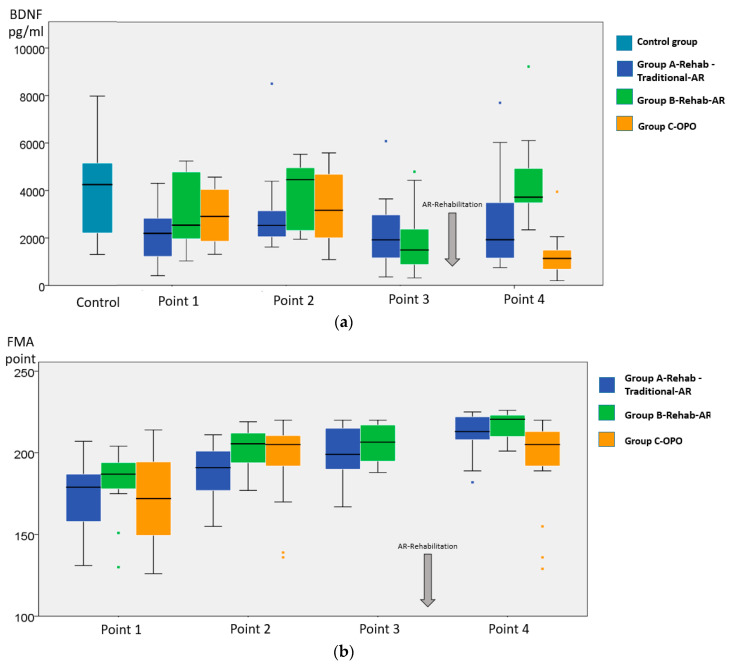
Serum brain-derived neurotrophic factor (BDNF) levels and Fugl–Meyer scale (FMA) scores at different points of observation. (**a**) Serum BDNF. **Comparisons**
**of groups between observation points:**
*A-Rehab-Traditional-augmented reality (AR):* P_1-2_ = **0.049** *; P_2-3_ = **0.010** *; P_3-4_ = **0.012** *; P_2-4_ = 0.476. *B-Rehab-AR:* P_1-2_ = **0.049**
*****; P_2-3_ = **0.016**
*****; P_3-4_ = **0.017**
*****; P_2-4_ = 0.374. *C-outpatient observation (OPO):* P_1-2_ < **0.001**
*****; P_2-4_ < **0.001 *. Comparisons between groups at observation points**: *Point 1:* P_A__-B_ = 0.21; P_B-C_ = 0.93; P_A-C_ = 0.08. *Point 2:* P_A-B_ = 0.10; P_B-C_ = 0.40; P_A-C_ = 0.43. *Point 3:* P_A-B_ = **0.049**
*****. *Point 4:* P_A-B_ = **0.02**
*****; P_B-C_ < **0.001**
*****; P_A-C_ = **0.02**
*****. *** Significant; *p* < 0.05**. (**b**). **FMA scores**. **Comparisons of groups between observation points:** A-Rehab-Traditional-AR: P_1-2_ < **0.001**
*****; P_2-3_ < **0.001**
*****; P_3-4_ < **0.001**
*****; P_2-4_ < **0.001**
*****. B-Rehab-AR: P_1-2_ < **0.001**
*****; P_2-3_ = 0.300; P_3-4_ < **0.001**
*****; P_2-4_ = **0.001**
*****. C-OPO: P_1-2_ = **0.018**
*****; P_2-4_ = 0.750. **Comparisons between groups at observation points**: Point 1: P_A-B_ = 0.81; P_B-C_ = 0.45; P_A-C_ = 0.95. Point 2: P_A-B_ = **0.01**
*****; P_B-C_ = 0.72; P_A-C_ = 0.12. Point 3: P_A-B_ = 0.31. Point 4: P_A-B_ = 0.45, P_B-C_ < **0.001**
*****, P_A-C_ = **0.02**
*****. *** Significant; *p* < 0.05.**

**Figure 2 brainsci-10-00623-f002:**
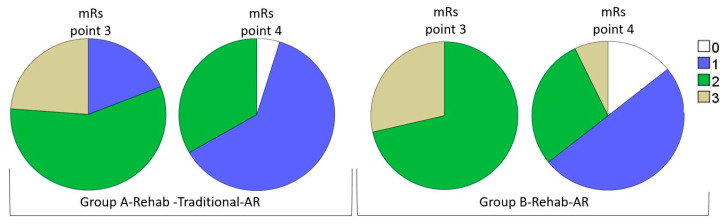
mRS scores during AR-based rehabilitation in Groups A and B. Comparisons of groups between observation points: A-Rehab-Traditional-AR: P_3-4_ < 0.001 *; B-Rehab-AR: P_3-4_ < 0.001 *. Comparisons between groups at observation points: mRS 3: P_A-B_
*=* 0.33; mRS 4: P_A-B_
*=* 0.93. * Significant; *p* < 0.05.

**Figure 3 brainsci-10-00623-f003:**
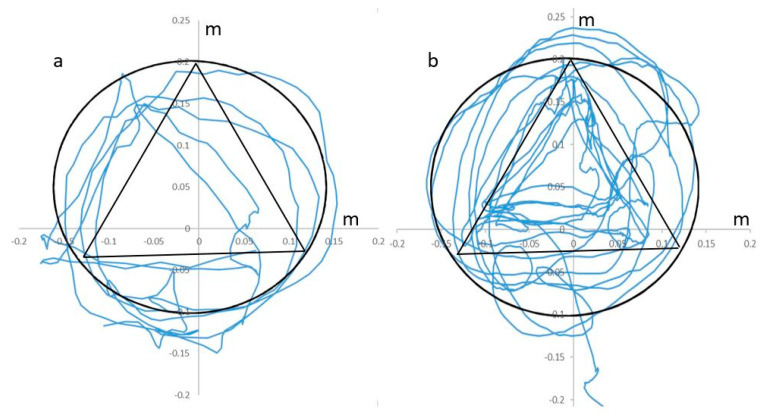
Trajectory of movement during upper extremity AR-Rehabilitation on the 1 and 10-day sessions (patient from A-Rehab-Traditional-AR group as an example); (**a**) 1st session and (**b**) 10th session. Trajectory of movement on X and Y coordinates by the left hand forefinger during execution of the Static domain task: black line—given trajectory, blue line—real trajectory of forefinger. Comparisons of movement parameters between 1st and 10th motor sessions: Variability of movements: P_1–10_ < 0.001 *; Number of completed tasks: P_1–10_ < 0.001 *; Maximum duration in one approach: < 0.001 *. * Significant; *p* < 0.05.

**Figure 4 brainsci-10-00623-f004:**
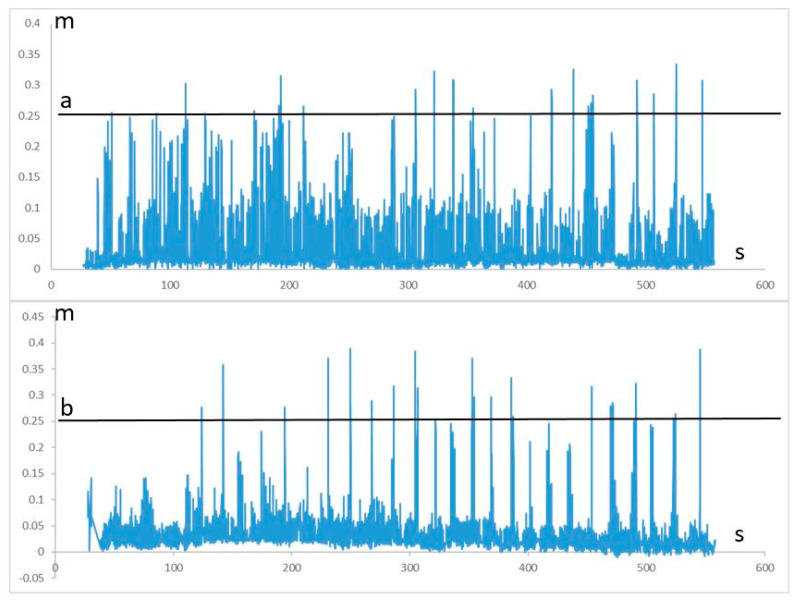
Trajectory of movement during AR-Rehabilitation on the 1 and 10-day sessions (patient from A-Rehab-Traditional-AR as an example); (**a**) 1st session and (**b**) 10th session. Height of raising the left foot during execution of the test of the Balance domain: black line—height of virtual obstacle, blue line—Y coordinate of left foot. Comparisons of movement parameters between 1st and 10th motor sessions: Height of raising the paretic leg: P_1–10_ = 0.003 *. Variance of the displacement for the central point: P_1–10_ < 0.001 *. * Significant; *p* < 0.05.

**Table 1 brainsci-10-00623-t001:** ΔBDNF in the patients groups.

ΔBDNF (pg/mL)	Ischemic Stroke *n* = 50			P_A-B_	P_B-C_	P_A-C_
	A-Rehab-Traditional-AR *n* = 21	B-Rehab-AR *n* = 14	C-OPO *n* = 15			
ΔBDNF day82−day14	−525 (−1073; 698)	−1231 (−1178; 2120)	−2415 (−3117; −760)	0.476	0.021 *	0.049 *
ΔBDNF day82−day45	513 (281; 1565)	2327 (924; 2529)	n/a	0.049 *	n/a	n/a
ΔBDNF day14−initial	572 (−307; 1521)	751 (182; 1020)	493 (147; 1093)	0.689	0.557	0.778

Data are shown as the median (Q1; Q3). P_A-B_—Probability A-Rehab-Traditional-AR vs. B-Rehab-AR; P_B-C_—Probability B-Rehab-AR vs. C-OPO; P_A-C_—Probability A-Rehab-Traditional-AR vs. C-OPO. * Significant; *p* < 0.05.

**Table 2 brainsci-10-00623-t002:** Serum BDNF in patients with ischemic stroke and control group.

Points	Serum BDNF (pg/mL)				P_A-Ctrl_	P_B-Ctrl_	P_C–Ctrl_
	A-Rehab-Traditional-AR *n* = 21	B-Rehab-AR *n* = 14	C-OPO *n* = 15	Control Group *n* = 50			
Point 1	2190 (1218; 2829)	2537 (1968; 4777)	2906.5 (1855; 4043)	4250 (2215; 5152)	0.001 *	0.295	0.086
Point 2	2525 (2050; 3144)	4460 (2317; 4958)	3164 (2002; 4686)	4250 (2215; 5152)	0.045 *	0.886	0.391
Point 3	1917 (1158; 2973)	1489 (877; 2366)	n/a	4250 (2215; 5152)	0.022 *	<0.001 *	n/a
Point 4	1923 (1149; 3488)	3719 (3485; 4929)	1131 (679; 1484)	4250 (2215; 5152)	0.012 *	0.693	<0.001 *

Data are shown as the median (Q1; Q3). P_A-Ctrl_—Probability A-Rehab-Traditional-AR vs. controls; P_B-Ctrl_—Probability B-Rehab-AR vs. controls; P_C-Ctrl_—Probability C-OPO vs. controls. * Significant; *p* < 0.05.

**Table 3 brainsci-10-00623-t003:** Serum BDNF in patients with ischemic stroke and control group.

A-Rehab-Traditional-AR *n* = 21	Point 1	Point 2	Point 3	Point 4	P_1-2_	P_2-3_	P_3-4_
FMA-Upper extremity	35 (31; 40)	42 (38; 50)	49 (43; 57)	61 (56; 64)	<0.001 *	<0.001 *	<0.001 *
FMA-Low extremity	24 (21; 27)	28 (24; 30)	29 (27; 33)	33 (29; 34)	0.002 *	0.001 *	0.001 *
FMA-Balance	10 (9; 12)	12 (10; 13)	12 (12; 14)	13 (12; 14)	0.003 *	0.06	0.002 *
**B-Rehab-AR** ***n* = 14**	**Point 1**	**Point 2**	**Point 3**	**Point 4**	**P_1-2_**	**P_2-3_**	**P_3-4_**
FMA-Upper extremity	39 (28; 45)	53 (49; 54)	53 (50; 57)	63 (58; 64)	<0.001 *	0.110	<0.001 *
FMA-Low extremity	26 (22; 28)	31 (26; 34)	32 (29; 34)	33 (29; 34)	0.010 *	0.070	0.080
FMA-Balance	12 (11; 12)	13 (12; 14)	13 (12; 14)	14 (13; 14)	0.010 *	0.320	0.040 *
**C-OPO** ***n* = 15**	**Point 1**	**Point 2**	**Point 3**	**Point 4**	**P_1-2_**	**P_2-3_**	**P_3-4_**
FMA-Upper extremity	39 (15; 45)	54 (47; 58)	N/A	54 (47; 59)	<0.001 *	N/A	N/A
FMA-Low extremity	24 (20; 29)	28 (27; 33)	N/A	29 (27; 33)	<0.001 *	N/A	N/A
FMA-Balance	11 (5; 13)	12 (12; 14)	N/A	12 (12; 14)	0.010 *	N/A	N/A

Data are shown as median (Q1; Q3). P_1-2_—Probability Point 1 vs. Point 2; P_2-3_—Probability Point 2 vs. Point 3; * Significant; *p* < 0.05.

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
