# Peer review of "Serum BDNF’s Role as a Biomarker for Motor Training in the Context of AR-Based Rehabilitation after Ischemic Stroke"

_brainsci, 2020, doi:10.3390/brainsci10090623_

Round 1
Reviewer 1 Report
The revised manuscript is significantly improved. Most of the questions have been corrected and well-answered.
Author Response
Thank you for your positive evaluation
Reviewer 2 Report
The manuscript's clarity is greatly improved.
Author Response
Thank you for your positive evaluation.
Reviewer 3 Report
Overall, the manuscript is much better than the first version. However, an important issue that must be addressed before accepting the manuscript is that the significance of the study is jeopardized because the study is mainly focused on the correlation between BDNF and treatment. From my point of view, it is more crucial to investigate/discuss the relationship between BDNF and clinical condition (e.g. motor improvement).
Please revise the format of the table base on the instruction (https://www.mdpi.com/journal/brainsci/instructions#).
Author Response
Overall, the manuscript is much better than the first version. However, an important issue that must be addressed before accepting the manuscript is that the significance of the study is jeopardized because the study is mainly focused on the correlation between BDNF and treatment. From my point of view, it is more crucial to investigate/discuss the relationship between BDNF and clinical condition (e.g. motor improvement).
Response: We thank the reviewer for this comment and for drawing our attention to a very interesting study by Qin et al. (2014). This allowed us to analyze the results of our research from a slightly different point of view. Therefore, we have added new data related to the analysis of the quality of movements during AR-rehabilitation and thus revealed the features of motor recovery. The question about the role of BDNF in motor recovery and adaptive behavior is undoubtedly debatable and open to date. But an experimental study in a mouse model by Qin and colleagues showed BDNF SNP-induced behavioral changes which may be reflected at the systemic level in stroke patients in the early recovery period. We were able to confirm this relationship by statistically significant changes in the parameters of movement quality. Thus, thanks to the reviewer's comment, we were able to combine and demonstrate clinical and laboratory data and parameters of movement quality, which collectively reflect the systems-level mechanism for enhanced motor recovery and role of BDNFVal66Met for functional recovery in early recovery of stroke.
We added:
In 2. Experimental Section
2.6 Clinical assessment of motor functions
Gait and movements of upper extremity changes during AR-rehabilitation were assessed using author parameters’ quality of movements:
- variability of movements when following a given trajectory;
- total number of completed movements (completed task) during one motor session;
- maximum duration of a tasks series in one approach without a rest;
- variance of the displacement for the central point of the body during the walk before crossing the obstacles;
- height of raising the leg on the affected side when stepping through virtual barrier.
The direct effects of AR-based rehabilitation were also evaluated by calculating the quality of movement during the first, fifth and tenth training sessions and interpreting detailed information obtained with parameters for assessing the quality of movements during the tasks in motor domains [33].
In 3. Results
Comparison of the movement parameters concerning the variability of movements during the 1 -10 training sessions did not reveal any significant differences between A-Rehab-Traditional-AR and B-Rehab-AR [33]. During AR-rehabilitation significant augmentation of the accuracy of following a given trajectory, improvement of reciprocal interaction of upper extremities muscles with a static-dynamic load and increased the muscle strength was observed (Figure 3). Patients also significantly increased the height of raising their leg on the affected side since 1st to 10th sessions of AR-rehabilitation (Figure 4).
Fig. 3 Trajectory of movement during upper extremity AR-Rehabilitation on 1st and 10th days sessions (patient from A-Rehab-Traditional-AR group as an example)
a -1st session and b - 10th session. Trajectory of movement on X and Y coordinates by the left hand forefinger during execution of the Static domain task: black line - given trajectory, blue line - real trajectory of forefinger.
Comparisons of movement parameters between 1st and 10th motor sessions:
- Variability of movements: P1-10 < 001*
- Number of completed tasks: P1-10 < 001*
- Maximum duration in one approach: < 001*
* Significant; P < 0.05.
Fig. 4 Trajectory of movement during AR-Rehabilitation on 1st and 10th days sessions (Patient from A-Rehab-Traditional-AR as an example)
a - 1st session and b - 10th session. Height of raising the left foot during execution of the test of the Balance domain: black line - height of virtual obstacle, blue line - Y coordinate of left foot.
Round 2
Reviewer 3 Report
The manuscript has been significantly improved. The authors added extra experiments (results) to answer my question. I don't have any concern to accept the conclusion of the manuscript.
This manuscript is a resubmission of an earlier submission. The following is a list of the peer review reports and author responses from that submission.
Round 1
Reviewer 1 Report
Comments:
The research manuscript of communication from Ekaterina et al. entitled “Serum BDNF reacts to AR-rehabilitation after ischemic stroke” described that ischemic stroke patients continuously received traditional rehabilitation and augmented reality rehabilitation (AR-rehabilitation) who significantly increased level of serum BDNF, especially during the AR-rehabilitation phase. The level of serum BDNF correlates to rehabilitation timing and intensity as well as recovery outcome has been well documented. Many previous studies suggested that serum BDNF is an indicator to evaluate treating efficacy and predicting disease outcome. However, this study combined regular rehabilitation and AR-rehabilitation significantly and stably increased serum BDNF. Authors concluded that fluctuated BDNF level of patients showed inconsistent improvement of motor activity.
Overall, applying the serum BDNF level to assess outcome and motor activity of ischemic stroke patients has been well studied. The new information of the manuscript is showing that AR-rehabilitation is able to maintain stably higher serum BDNF level. It is a predictable, not exciting, finding.
There are couples of crucial issues that need to be aware, which are as following:
- One way or the other, the title does not clearly reflect the theme of the manuscript,
- The description of “Results” and “Conclusion” sections are hardly to follow. Reviewer could not catch the theme of the manuscript. The written English of the manuscript needs being extensively improved.
- The figure and table legends are too simple to understand. A good figure legend is brief and straightforward to understand without going back to main text of the “Results” section.
- The expression statistical significance is commonly using symbol * or # to label, but not red colored in the text or the tables.
- Request peer researchers and senior researchers to go through the manuscript and get feedback before resubmission.
- Highly recommend sending the manuscript to English editing professionals before resubmission.
Minor points
- In the line 56 and 210, the word “larger” to describe comparing levels of BDNF should be replaced to “higher”.
- In the line 60-61, “…their participation in public processes.”, commonly written as “…their participation in social activities.”
- Please rephrase to simple and clear sentences. For example, the first sentence of “Results” could be rewritten as “Fifty ischemic stroke patients and equal number of healthy individuals were recruited in this study”.
- In line 213, the word “observed” should be commonly using “examined”.
- The font of chart label is relatively too small.
Author Response
We thank all Reviewers for their positive evaluation of our study and helpful criticisms and suggestions, following which we significantly modified our manuscript. We believe that these changes have significantly improved our manuscript and clarified our data presentation. Below, please find our response to specific comments made by first Reviewer.
The research manuscript of communication from Ekaterina et al. entitled “Serum BDNF reacts to AR-rehabilitation after ischemic stroke” described that ischemic stroke patients continuously received traditional rehabilitation and augmented reality rehabilitation (AR-rehabilitation) who significantly increased level of serum BDNF, especially during the AR-rehabilitation phase. The level of serum BDNF correlates to rehabilitation timing and intensity as well as recovery outcome has been well documented. Many previous studies suggested that serum BDNF is an indicator to evaluate treating efficacy and predicting disease outcome. However, this study combined regular rehabilitation and AR-rehabilitation significantly and stably increased serum BDNF. Authors concluded that fluctuated BDNF level of patients showed inconsistent improvement of motor activity.
Overall, applying the serum BDNF level to assess outcome and motor activity of ischemic stroke patients has been well studied. The new information of the manuscript is showing that AR-rehabilitation is able to maintain stably higher serum BDNF level. It is a predictable, not exciting, finding.
We thank Reviewer #1 for his/her valuable comments. We have changed the title of our paper and have completely rewritten the results section (and changed the data presentation in tables and figures). Thereafter we have submitted the revised manuscript to the MDPI English editing service for plagiarism check and language correction. We believe that we have addressed the comments of highly esteemed reviewer sufficiently.
There are couples of crucial issues that need to be aware, which are as following:
- One way or the other, the title does not clearly reflect the theme of the manuscript,
Response: We thank the reviewer for pointing this out. We changed the title: Serum BDNF’s Role as a Biomarker for Motor Recovery in the Context of AR-Based Rehabilitation after Ischemic Stroke.
- The description of “Results” and “Conclusion” sections are hardly to follow. Reviewer could not catch the theme of the manuscript. The written English of the manuscript needs being extensively improved.
Response: We thank the reviewer for this comment. Motivated by the reviewer we have rewritten the results section and have edited the conclusions.
- The figure and table legends are too simple to understand. A good figure legend is brief and straightforward to understand without going back to main text of the “Results” section.
Response: We thank the reviewer for this comment. Motivated by the reviewer we have changed the tables and figures and rephrased their legends. Moreover these legends have been edited as suggested by the MDPI language service.
- The expression statistical significance is commonly using symbol * or # to label, but not red colored in the text or the tables.
Response: We thank the reviewer for this comment. We have marked significant difference with * and printed the figure in bold for clarity.
- Request peer researchers and senior researchers to go through the manuscript and get feedback before resubmission.
Response: We thank the reviewer for this suggestion. Motivated by the reviewer we have rewritten the results section and have edited the introduction and discussion.
- Highly recommend sending the manuscript to English editing professionals before resubmission.
Response: We thank the reviewer for this suggestion.
Minor points
- In the line 56 and 210, the word “larger” to describe comparing levels of BDNF should be replaced to “higher”.
Response: Thank you for this comment. We have changed this.
- In the line 60-61, “…their participation in public processes.”, commonly written as “…their participation in social activities.”
Response: Thank you for this comment. We changed this sentence accordingly: Stroke causes motor-function deficits that significantly reduce the patient’s mobility, activities of daily living (ADL) and participation in social activities, resulting in a decreased quality of life (QoL) [5].
- Please rephrase to simple and clear sentences. For example, the first sentence of “Results” could be rewritten as “Fifty ischemic stroke patients and equal number of healthy individuals were recruited in this study”.
Response: Thank you for this comment. We changed this sentence accordingly.
- In line 213, the word “observed” should be commonly using “examined”.
Response: Thank you for this comment. We have changed this.
- The font of chart label is relatively too small.
Response: Thank you for this comment. We have changed the figures.
Reviewer 2 Report
The manuscript from Koroleva et al is interesting and provides correlational evidence for the use of AR- rehabilitation following ischemia based on BDNF shifts. There are however, some finer points that could benefit from clarification.
- Using Group A, B, C without a graphic to explain the schemes is confusing and makes the data harder to interpret. It would help to clearly provide a timeline or similar to demonstrate visually how each group is different. The bullet points in 2.3 are insufficient. In addition, please label Group A, B, and C with descriptive names like Rehab-AR, AR, vs no AR, etc so that data are easier to interpret.
- The figures are not clearly labelled or organized. For instance, I assume “Pa” is p value but that is not stated anywhere in the table or in the legend. The p values for the BDNF graphs are just above the graph not in a detailed figure legend (which is also missing), so it is difficult to assign which p value goes with what graph.
- Please note what form of BDNF xMAP® Technology measures (all BDNF, mature, prepro BNDF), as that will change the interpretation of the results.
- The introduction and discussion would benefit from the addition of literature about BDNF and TrkB signaling in vasculature.
Author Response
We thank all Reviewers for their positive evaluation of our study and helpful criticisms and suggestions, following which we significantly modified our manuscript. We believe that these changes have significantly improved our manuscript and clarified our data presentation. Below, please find our response to specific comments made by the second Reviewer.
The manuscript from Koroleva et al is interesting and provides correlational evidence for the use of AR- rehabilitation following ischemia based on BDNF shifts. There are however, some finer points that could benefit from clarification.
We thank Reviewer #1 for his/her valuable comments which helped to considerably improve our manuscript and data presentation.
Using Group A, B, C without a graphic to explain the schemes is confusing and makes the data harder to interpret. It would help to clearly provide a timeline or similar to demonstrate visually how each group is different. The bullet points in 2.3 are insufficient. In addition, please label Group A, B, and C with descriptive names like Rehab-AR, AR, vs no AR, etc so that data are easier to interpret.
Response: We thank the reviewer for this comment. Motivated by the reviewer we have added a figure depicting the study design as Appendix A. We renamed the studied patient groups as: A-Rehab-Traditional-AR, B-Rehab-AR, C-OPO.
The figures are not clearly labelled or organized. For instance, I assume “Pa” is p value but that is not stated anywhere in the table or in the legend. The p values for the BDNF graphs are just above the graph not in a detailed figure legend (which is also missing), so it is difficult to assign which p value goes with what graph.
Response: We thank the reviewer for this comment. We have changed both tables and figures. We explain the meaning of the probability symbols in the legend of the tables.
Please note what form of BDNF xMAP® Technology measures (all BDNF, mature, prepro BNDF), as that will change the interpretation of the results.
Response: We thank the reviewer for this comment. We have specified in the text that we only measured mature BDNF.
The introduction and discussion would benefit from the addition of literature about BDNF and TrkB signalling in vasculature.
Response: We thank the reviewer for this comment. We have added 2 references:
- Bathina, S., Das, U.N. Brain-derived neurotrophic factor and its clinical implications. Arch Med Sci 2015, 11(6), 1164-1178. DOI: 10.5114/aoms.2015.56342
- Benarroch, E.E. Brain derived neurotrophic factor: Regulation, effects, and potential clinical relevance. Neurology. 2015, 84, 1693–1704. DOI: 10.1212/wnl.0000000000001507
Reviewer 3 Report
General comments:
The authors try to investigate the role of serum BDNF in the rehabilitation process from ischemic stroke. They report BDNF seems to react during active AR-rehabilitation treatment. However, the no-treatment control group shows gradually decreased BNDF after the acute phase of ischemic stroke. In addition, since serum BDNF level is independent of the clinical condition of the patients, the authors propose that the BDNF level is related to the consolidation of the learned motor skills.
In my opinion, it is more important to investigate/discuss the relationship between the neurotrophic factors and clinical condition (e.g. motor improvement), rather than focus on the correlation between BNDF and treatment. Such as previous report (e.g. Luye Qin, et. al. 2014 ) suggests BDNF level related to motor recovery.
Experiment Section and Results:
The authors use a wide range of methods to reach their conclusions. It would be helpful for the reader to have an organized methods section in supplementary material with more details with references. I also suggest moving non-essential information (e.g. Table 1, Appendix A, et. al.) to supplementary material.
It is good to see the discussion from the authors upon the challenges and potential caveats of the study (Line 420~425).
Presentation:
Overall the manuscript is easy to follow, it still has some room for improvement.
- Please revise the affiliations base on the instructions (https://www.mdpi.com/journal/brainsci/instructions#).
- Please use the Microsoft Word template or LaTeX template to prepare your manuscript, especially the format of the tables (https://www.mdpi.com/journal/brainsci/instructions#).
- Please use high-resolution figures and increase the text size in Figures (line 227, line 232, line 238, line 242).
- I would like to add a * after P if P-value is smaller than 0.05, rather indicate it in red.
Author Response
We thank all Reviewers for their positive evaluation of our study and helpful criticisms and suggestions, following which we significantly modified our manuscript. We believe that these changes have significantly improved our manuscript and clarified our data presentation. Below, please find our response to specific comments made by the third Reviewer.
- General comments:
The authors try to investigate the role of serum BDNF in the rehabilitation process from ischemic stroke. They report BDNF seems to react during active AR-rehabilitation treatment. However, the no-treatment control group shows gradually decreased BNDF after the acute phase of ischemic stroke. In addition, since serum BDNF level is independent of the clinical condition of the patients, the authors propose that the BDNF level is related to the consolidation of the learned motor skills.
We thank Reviewer #1 for his/her valuable comments which helped to considerably improve our manuscript and data presentation.
In my opinion, it is more important to investigate/discuss the relationship between the neurotrophic factors and clinical condition (e.g. motor improvement), rather than focus on the correlation between BNDF and treatment. Such as previous report (e.g. Luye Qin, et. al. 2014 ) suggests BDNF level related to motor recovery.
Response: We thank the reviewer for this comment. However, our results indicate that BDNF levels are more related to training motor function than to the improvement of motor function which results from such training. We apologize to have missed your suggestion to cite the study of Qin et al.
- Experiment Section and Results:
The authors use a wide range of methods to reach their conclusions. It would be helpful for the reader to have an organized methods section in supplementary material with more details with references. I also suggest moving non-essential information (e.g. Table 1, Appendix A, et. al.) to supplementary material.
Response: We thank the reviewer for this comment. We added a figure depicting the study design as Appendix A and described the rehabilitation methods in Appendix B, C, and D. We transferred table 1 to Appendix E.
It is good to see the discussion from the authors upon the challenges and potential caveats of the study (Line 420~425).
Response: We greatly appreciate this positive statement.
- Presentation:
Overall the manuscript is easy to follow, it still has some room for improvement.
We have changed the title of our paper and have completely rewritten the results section (and changed the data presentation in tables and figures). Thereafter we have submitted the revised manuscript to the MDPI English editing service for plagiarism check and language correction. We believe that all Reviewers remarks have been addressed in the current revision of our paper.
- Please revise the affiliations base on the instructions (https://www.mdpi.com/journal/brainsci/instructions#).
- Please use the Microsoft Word template or LaTeX template to prepare your manuscript, especially the format of the tables (https://www.mdpi.com/journal/brainsci/instructions#).
- Please use high-resolution figures and increase the text size in Figures (line 227, line 232, line 238, line 242).
I would like to add a * after P if P-value is smaller than 0.05, rather indicate it in red.